# Comparative Study of Temperature Impact in Spin-Torque Switched Perpendicular and Easy-Cone MTJs

**DOI:** 10.3390/nano13020337

**Published:** 2023-01-13

**Authors:** Jingwei Long, Qi Hu, Zhengping Yuan, Yunsen Zhang, Yue Xin, Jie Ren, Bowen Dong, Gengfei Li, Yumeng Yang, Huihui Li, Zhifeng Zhu

**Affiliations:** 1School of Information Science and Technology, Shanghai Tech University, Shanghai 201210, China; 2Beijing Superstring Academy of Memory Technology, Beijing 100176, China; 3Shanghai Engineering Research Center of Energy Efficient and Custom AI IC, Shanghai 201210, China

**Keywords:** spin-transfer torque, easy-cone magnetization, precession switching, current overdrive, write error rate

## Abstract

The writing performance of the easy-cone magnetic tunnel junction (MTJ) and perpendicularly magnetized MTJ (pMTJ) under various temperatures was investigated based on the macrospin model. When the temperature is changed from 273 K to 373 K, the switching current density of the pMTJ changes by 56%, whereas this value is only 8% in the easy-cone MTJ. Similarly, the temperature-induced variation of the switching delay is more significant in the pMTJ. This indicates that the easy-cone MTJ has a more stable writing performance under temperature variations, resulting in a wider operating temperature range. In addition, these two types of MTJs exhibit opposite temperature dependence in the current overdrive and write error rate. In the easy cone MTJ, these two performance metrics will reduce as temperature is increased. The results shown in this work demonstrate that the easy-cone MTJ is more suitable to work at high temperatures compared with the pMTJ. Our work provides a guidance for the design of STT-MRAM that is required to operate at high temperatures.

## 1. Introduction

Spin-transfer torque magnetic random-access memory (STT-MRAM) with perpendicular magnetization is a promising and emerging technology. Benefiting from its nonvolatility, faster access speed compared with DRAM and higher capacity compared with SRAM, STT-MRAM has become a competitive candidate to replace SRAM and DRAM to realize the universal memory [1,2]. STT-MRAM consists of the transistor and magnetic tunnel junction (MTJ). As shown in Figure 1a, MTJ consists of a nonmagnetic spacer layer and two ferromagnetic layers. One of the ferromagnetic layers is magnetically pinned, named the pinned layer (PL). The magnetization of the other ferromagnetic layer can be changed by the external excitations. This layer is named the free layer (FL). Since the two magnetic layers have the same easy axis, their magnetization, m_FL_ and m_PL_, can only be parallel or antiparallel. According to the tunneling magnetoresistance effect, the electrical resistance of MTJ is low (high) when m_FL_ and m_PL_ are parallel (antiparallel), which can be used to represent the binary states [3].

In the STT-MRAM, a current flowing through the device generates the STT (indicated by the blue arrows in Figure 1a), which can be used to switch the magnetization of the FL. The magnitude of STT is proportional to the relative angle between m_FL_ and m_PL_ (*θ*), given as *τ*_STT_ m_FL_ × (m_FL_ × m_PL_), where *τ*_STT_ is the coefficient dependent on the physical parameters of MTJ [4]. Restricted to the collinear magnetization at equilibrium, the STT is vanished although the current is applied. In consequence, the STT-induced magnetization switching relies on the thermal fluctuations that can break the collinearity of m_FL_ and m_PL_. Nevertheless, since these fluctuations are random in nature, the switching time is greatly varied and cannot be controlled [5,6,7]. This stochasticity put forward stringent requirements for the design of the device to achieve an appropriate write error rate [8,9,10]. Some studies focused on the design of the device structure to increase the relative angle between m_FL_ and m_PL_. For example, the PL with a tilted easy axis has been employed, but it brought difficulties to the epitaxial growth. Two spin polarizing layers with in-plane and out-of-plane magnetization are also considered. However, it is limited by the complex stacking structure [11,12,13,14].

Some analytical and macrospin studies have revealed that the switching characteristics of MTJs can be improved by exploiting a conically magnetized FL [15,16,17,18], which can be realized by the second-order magnetic anisotropy. As shown in the right panel of Figure 1a, when the demagnetizing energy is partially or fully balanced by the first-order anisotropy energy, the second-order magnetic anisotropy leads to an easy cone state which has a fixed polar angle (*θ*_c_) with respect to the out-of-plane direction [19,20,21,22]. This self-contained misalignment brings advantages such as faster switching, lower switching current density, and better write error rate compared with the pMTJ. However, almost all studies that compare the easy-cone MTJ and pMTJ focus on the room temperature performance. Meanwhile, the temperature studies on the easy-cone MTJ only focus on how to stabilize the easy-cone magnetization under various temperatures or the temperature effect on the TMR ratio [15,19]. The temperature effect on the writing performance of the easy-cone MTJ has not been systematically investigated. It has been reported that there are many issues in the pMTJ as the temperature is varied. For example, at low temperatures, the reduced thermal fluctuations decrease the relative angle between m_FL_ and m_PL_. The initial STT will thus be decreased and the incubation delay will be increased [23]. However, in the easy-cone MTJ, this angle is mainly determined by the magnetocrystalline anisotropy energy and demagnetizing energy. It will not be constrained by thermal fluctuations. At high temperatures, the thermal stability (Δ) of the pMTJ will be reduced a lot [24]. However, the existence of second-order anisotropy in the easy-cone MTJ increases the energy barrier between the equilibrium magnetization state and the in-plane magnetization state (*E*_B_), which compensates for the reduction in Δ. Δ is closely related to the writing performance of the MTJ. Therefore, it is believed that the writing performance of the easy-cone MTJ is more stable under temperature variations. This makes it more suitable to operate at extreme temperatures.

In this work, the width and length of FL is assumed to be 20 nm and 48 nm, respectively. Previous studies pointed out that the ferromagnets with a lateral size smaller than 80 nm can be described by the macrospin model [25]. Thus, we use the macrospin model to simulate the dynamics of the device. Based on the temperature dependence of physical parameters, we show that the easy-cone MTJ has a larger Δ at high temperatures due to the second-order anisotropy. Meanwhile, Δ changes less significantly in the easy-cone MTJ as the temperature is varied. As a result, the writing performance of the easy-cone MTJ show a better immunity under temperature variations. In addition, our results show that the easy-cone MTJ exhibits a smaller switching current density and a lower write error rate at high temperatures. These results further confirm the remarkable potential of the easy-cone MTJ in memory applications. Meanwhile, our work can stimulate the design of high performance STT-MRAM operating at high temperatures.

## 2. Methodology

In the easy-cone state, the tilted magnetization is stabilized by the competition of the magnetic anisotropy energy and demagnetization energy. Here, the former consists of both first- and second-order magnetic anisotropy energies. Thus, the energy density of easy-cone FL is given by:(1)$$\varepsilon =\frac{1}{2}\mu_{0}M_{sat}^{2}(N_{xx}m_{x}^{2}+N_{yy}m_{y}^{2}+N_{zz}m_{z}^{2})+K_{u1}(1-m_{z}^{2})+K_{u2}{\left(1-m_{z}^{2}\right)}^{2}$$

Here, *N_xx_*, *N_yy,_* and *N_zz_* represent demagnetization coefficients in the three dimensions. μ_0_ is the vacuum permeability and *M_sat_* is the saturation magnetization, which refers to the maximum magnetization that the ferromagnet can reach when it is magnetized by a magnetic field. *K_u_*_1_ and *K_u_*_2_ are the first- and second-order magnetic anisotropy constants, respectively. The demagnetization energy will generate an in-plane shape anisotropy field, **H**_IP_, with the magnitude of *M*_sat_ (*N_zz_*−*N_xx_*) [26]. Thus, the energy density of FL can be rewritten as:(2)$$\varepsilon =K_{u1,eff}\left(1-m_{z}^{2}\right)+K_{u2}{\left(1-m_{z}^{2}\right)}^{2}$$
where *K_u_*_1*,eff*_ is the effective first-order anisotropy constant with demagnetization energy considered, given by *K_u_*_1*,eff*_ = *K_u_*_1_−(1/2)μ_0_*M*_sat_^2^(*N*_zz_−*N*_xx_). Figure 1b reveals that the competition of *K_u_*_1*,eff*_ and *K_u_*_2_ determines the *θ*_c_. When *K_u_*_1*,eff*_ < 0 and −*K_u_*_2_/*K_u_*_1*,eff*_ > 1/2 (indicated by the shaded region), the easy-cone state can be stabilized. By minimizing the energy density of the FL, the equilibrium polar angle can be expressed as:(3)$$\theta_{c}=\sin^{-1}\sqrt{\frac{-K_{u1,eff}}{2K_{u2}}}$$

For practical applications, *θ*_c_ is often required to be smaller than 15° to ensure an appropriate Δ.

In this study, the temperature dependence of physical parameters is included. Here, we mainly consider the dependence of *M_sat_*, *K_u_*_1_, and the spin polarization factor *P* on temperature. They can be modeled by following formulas [27,28,29]:(4)$$M_{sat}(T)=M_{sat}(0)[1-{\left(\frac{T}{T_{c}}\right)}^{3/2}]$$
(5)$$\;\;K_{u1}\left(T\right)=K\left(0\right){\left[\frac{M_{sat}\left(T\right)}{M_{sat}\left(0\right)}\right]}^{3}$$
(6)$$P\left(T\right)=P\left(0\right)\left(1-\beta T^{\frac{3}{2}}\right)$$
where *T* is the temperature, and *β* = 2 × 10^−5^ K^−3/2^ is the fitting parameters dependent on materials. *T*_c_ = 750 K is the Curie temperature. *M_sat_*(0), *K_u_*_1_(0), and *P*(0) are the values at *T* = 0 K. Their values are given in Table 1 so that the corresponding values at *T* = 300 K are consistent with those reported in [30]. The pMTJ and easy-cone MTJ studied in this work differ only in magnetic anisotropy energy. That is, *M_sat_*(0) and *P*(0) of both devices are the same but the *K_u_*_1_(0) is different. As for *K_u_*_2_, previous studies pointed out that the significant *K_u_*_2_ does not intrinsically originate from the ferromagnetic interfaces. The spatial fluctuations of film thickness and atomic structure at the interface should be responsible for its emergence [31,32,33]. Therefore, we exclude the temperature dependence of *K_u_*_2_ in this study. As illustrated in Figure 2a–c, all these three parameters decrease as *T* is increased. The same monotonicity of *M_sat_* and *K_u_*_1_ makes it difficult to directly derive the changes of Δ as the temperature is varied, and we resort to the numerical calculation. The Δ of the easy-cone state is given by [14]:(7)$$∆=\frac{E_{B}}{k_{B}T}=\frac{\left(\varepsilon \left(\theta =\frac{\pi }{2}\right)-\varepsilon \left(\theta =\theta_{c}\right)\right)V}{k_{B}T}$$

Here, *V* is the volume of FL and *k_B_* is the Boltzmann constant. Figure 2d shows Δ as a function of *T* for the pMTJ and easy-cone MTJ. It is important to note that we set Δ = 60 at room temperature (*T* = 300 K) in both devices for a fair comparison between them. It can be observed that the pMTJ has a higher Δ before the intersection point due to the larger *K_u_*_1_(0). However, it becomes more thermally unstable than the easy-cone MTJ when *T* is increased. We attribute this to the presence of *K_u_*_2_ in the easy-cone MTJ and the faster decreasing *K_u_*_1_ in the pMTJ. As *T* is increased, the invariant *K_u_*_2_ in the easy-cone MTJ can produce a stable *E_B_* to slow down the degradation of Δ. In contrast, the rapidly decreased *K_u_*_1_ in the pMTJ leads to a sharply reduced *E_B_*. Thus, under high-temperature circumstance, easy-cone MTJ has a superior Δ. This indicates that the easy-cone MTJs have a better performance in the data retention when they are operating at elevated temperatures.

The current driven magnetization dynamics is studied by solving the Landau–Lifshitz–Gilbert–Slonczewski (LLGS) equation d**m**/d*t* = −γ**m** × **H**_eff_ + *α***m** × d**m**/d*t* − *γη*ℏ*J*/(2 *et*_FL_*M*_sat_)**m** × (**m** × **σ**_STT_) [34] with the gyromagnetic *γ*, the current density *J*, the reduced Planck constant ℏ, the electron charge *e*, the thickness of FL, *t*_FL_ = 1.2 nm, the damping constant, *α* = 0.01, the spin polarization, **σ**_STT_, and the effective magnetic field, **H**_eff_, which includes the first- and second-order magnetic anisotropy field, the demagnetizing field, and the thermal field. The STT efficiency *η = P*/[1 + *P*^2^cos(*θ*)] is determined by the spin polarization factor and polar angle of FL magnetization.

## 3. Results and Discussion

To study the temperature effect, we consider the standard operating temperature range of commercial electronic devices, from 273 K to 343 K. To further verify the superiority of the easy-cone MTJ at higher temperatures, we also perform the simulations at the temperature range of 343–373 K. The results show that the easy-cone MTJ is still superior. Therefore, it can be predicted that the advantages of the easy-cone MTJ will become even more apparent above 373 K. When the temperature is smaller than 273 K, it is hard to obtain the easy-cone state since the first-order anisotropy constant is so large that the anisotropy energy cannot be compensated by the demagnetizing energy. Finally, we determined the temperature region 273–373 K. We consider that the temperature affects the writing performance in two ways. One is the temperature dependence of physical parameters, which dominates the variations of the device performance. The other is the influence of the temperature-dependent **H**_thermal_, which arises from the thermal fluctuations. To investigate the intrinsic writing performance, we firstly exclude the **H**_thermal_. To determine the equilibrium state for the easy-cone MTJ, *θ*_c_ at various temperatures is calculated using Equation (3), which is shown in Figure 3a. Meanwhile, we determine the *θ*_c_ at *T* = 273 K to *T* = 373 K using the numerical simulation. The results of the calculation and simulation fit well. It can be seen that the misalignment between the FL and PL will become more pronounced at a higher *T*. The maximum *θ*_c_ in our simulation reaches 19° at *T* = 373 K. It has been demonstrated that a larger *θ*_c_ will lead to lower intrinsic switching current density (*J*_sw0_) and switching delay (*t*_sw_). *J*_sw0_ is defined as the critical current density above which STT can overcome the damping and magnetization switching will be initialized. A small *J*_sw0_ should be promised to ensure a low-power consumption. For the easy-cone MTJ and pMTJ, *J*_sw0_ is given by Equations (8) and (9), respectively [14,35].
(8)$$J_{\mathrm{sw}0,\mathrm{easy}-\mathrm{cone}}=\frac{8}{3\sqrt{6}}\frac{\alpha t_{FL}\left|e\right|}{\hbar P}\sqrt{\frac{{\left(K_{u1,eff}+2K_{u2}\right)}^{3}}{K_{u2}}}$$
(9)$$\;J_{\mathrm{sw}0,\mathrm{PMA}}=\frac{4\alpha t_{FL}\left|e\right|}{\hbar P}K_{u1,eff}$$

As depicted in Figure 3b, *J*_sw0_ of the easy-cone MTJ and pMTJ will drop as *T* is increased. In our study, the spin polarization *P* and *K*_u1,eff_ determine *J*_sw0_ when *T* varies. The reduced *K*_u1,eff_ at high temperatures leads to a lower Δ, and consequently a smaller current density is able to realize magnetization switching. In contrast, the reduced *P* results in a weaker STT, which calls for additional current density to switch the magnetization [36,37,38]. Note that *K*_u1,eff_ is more sensitive to temperatures, it is the dominated parameter in the variation of *J*_sw0_. Thus, the *J*_sw0_ curves of both easy-cone MTJ and pMTJ show a downward trend. It is worth noting that the decrease in *J*_sw0_ in pMTJ is steep, while that of the easy-cone MTJ shows a much gradual trend. This is because the variation of *K*_u1,eff_ in the pMTJ is more significant. Therefore, compared with the easy-cone MTJ, the pMTJ has a much lower *J*_sw0_ when *T* is higher than 323 K. However, this is at the cost of its stability under temperature variation. For example, in the range of 273 K to 373 K, *J*_sw0_ of the pMTJ changes by 56%. This value in the easy-cone MTJ is only 8%. In practical applications, this stability is very attractive since it can promise a wider operating temperature range.

Next, we examined the switching delay *t*_sw_, which characterizes the writing speed of STT-MRAM. Here, *t*_sw_ is defined as the required time for the magnetic moment switching from *θ* = *θ*_c_ to *θ* = 90°. As illustrated in Figure 3c,d, *t*_sw_ of both MTJs decreases as *T* is increased. This stems from the reduced *M*_sat_ at a high *T*, which can enhance the STT. In particular, for the easy-cone MTJ, the initial *θ*_c_ becomes larger with increasing *T*, resulting in a larger STT [17]. Such an increase in STT has been confirmed to accelerate the magnetic switching [14]. It is observed that the same as *J*_sw0_, for the easy-cone MTJ, the variation of *t*_sw_ as a function of temperatures is less significant compared with that in the pMTJ. This is more pronounced at moderate current densities 6 × 10^10^ to 8 × 10^10^ A/m^2^.

At finite temperatures, the thermal fluctuations cannot be ignored, which leads to a stochastic switching. The thermal fluctuations are included as an effective random field:(10)$$H_{thermal}=\zeta \sqrt{\frac{2k_{B}T\alpha }{VM_{sat}\left(1+\alpha^{2}\right)\delta t}}$$
where ζ is a vector with three components that are independent Gaussian random variables, and the time-step δt is 10 ps in our simulation. We then investigate the writing performance of both types of MTJs with pulse duration *τ* = 1 ns, 2 ns, and 5 ns. When *τ* < 5 ns, the switching in the conventional pMTJ is in the fast precession regime, where a large current is required for the successful switching [5,6]. As a result, the torque from thermal fluctuations plays a negligible role in the magnetization switching. However, it is important to note that the thermal fluctuations also affect the distribution of initial magnetization, which has a strong impact in the switching current density. Therefore, before studying the switching dynamics, one has to determine *θ* by taking the thermal fluctuations into account. To capture the thermal distribution of the magnetization, we have solved the LLGS equation with the effective thermal field described by Equation (10). Figure 4a shows the mean of *θ* with thermal fluctuations considered (*θ*_thermal_) as a function of temperatures for the easy-cone MTJ and pMTJ. It can be observed that raising *T* helps achieve a higher *θ*_thermal_ for both devices, leading to easier spin transfer switching. In addition, *θ*_thermal_ of the easy-cone MTJ changes more violently with *T*. However, thermal fluctuations play different roles in the variations of *θ*_thermal_ in the two devices. In the pMTJ, a larger *T* results in an intense thermal fluctuation, and further enhances the precession of the FL magnetization. Therefore, *θ*_thermal_ will increase accordingly. In contrast, for the easy-cone MTJ, the variation of *θ*_thermal_ mainly arose from the temperature dependence of the physical parameters. As illustrated in Figure 4b, at high temperatures (*T* > 298 K), the difference between *θ*_thermal_ and *θ*_c_, defined as Δ*θ*, is within 1°. However, at low temperature *T* = 273 K, *θ*_thermal_ was enlarged a lot with respect to the intrinsic *θ*_c_. This is because the effective anisotropy field in the easy-cone MTJ is decreased at low temperatures, which becomes comparable to **H**_thermal_. Three components of the magnetization tend to have the same statistical value under the influence of **H**_thermal_. Thus, thermal fluctuations can help achieve a higher *θ*_thermal_.

In the fast precession regime, the required current density for the magnetization switching is several times of *J*_sw0_. We define *J*_sw_ as the current density, at which the switching probability is 50% and investigate how the current overdrive *J*_sw_/*J*_sw0_ is affected when *T* is varied. The theoretical expression of the overdrive is 1 + [ln(π/2*θ*)/*J*_sw0_*τ*], which is determined by the combined effect of *θ* and *J*_sw0_ [5]. With increased *T*, benefiting from the stronger thermal fluctuations and reduced *K*_u1,eff_, *θ* will be enlarged, leading to a smaller overdrive. However, this will be compensated by the reduced *J*_sw0_ at higher *T* (see Figure 3b). It can be observed in Figure 4c,d that the overdrive of the two devices has an opposite temperature dependence. For the pMTJ, it shows an upward tendency as *T* is increased, while that of the easy-cone MTJ decreases at elevated *T*. This is attributed to the larger variations of *J*_sw0_ compared with the change of *θ* in the pMTJ. Therefore, *J*_sw0_ dominates the variations of the overdrive. However, for the easy-cone MTJ, since *J*_sw0_ barely changes as *T* is varied, *θ* dominates the variations of the current overdrive. As mentioned above, *θ* will be enlarged at higher *T*. Therefore, the overdrive will decrease as *T* is raised. In both devices, the variations of the overdrive mainly arise from the temperature dependence of the physical parameters. In contrast, the thermal fluctuations are less important in the fast precession regime. However, when *τ* is 5 ns, the overdrive of the easy-cone MTJ is almost unchanged as *T* is varied. It is confirmed that the easy-cone MTJ enters dynamic reversal regime, which is a transitory stage of the fast precession and thermally activated regimes. In this regime, restricted to the short *τ*, the reduction in overdrive is more gradual. Only when *τ* is further increased, and the magnetization switching enters thermally activated regime, the overdrive can be further decreased and becomes less than 1 [39]. In conclusion, in the precession regime, the easy-cone MTJ requires lower current density to realize switching than the pMTJ at high temperatures.

The write error rate is the probability of non-switching cases with current applied. In order to ensure the data writing function, the write error rate should be low enough. Figure 5 shows the write error rate as a function of the current density *J* normalized by *J*_sw0_ at different *τ*. An opposite temperature dependence of WER can be observed for the pMTJ and easy-cone MTJ. For the pMTJ, it requires a larger *J*/*J*_sw0_ to ensure an appropriate write error rate as *T* is raised, resulting in the right shift of the write error rate curves. In contrast, for the easy-cone MTJ, the *J*/*J*_sw0_ required to achieve the same write error rate becomes smaller at elevated *T*. For instance, in Figure 5a, the pulse duration is 2 ns and both types of MTJs work in the precession regime. To realize a write error rate of 10^−3^ in the easy-cone MTJ, the required *J*/*J*_sw0_ is 2.67 at *T* = 373 K, and this value is increased to 3.18 at a lower *T* = 273 K. However, in the pMTJ, the corresponding *J*/*J*_sw0_ is 6.31 at *T* = 373 K and decreased to 3.62 at *T* = 273 K. It can be observed in Figure 5b that when *T* changes from 273 K to 373 K, the required *J* for a write error rate of 10^−3^ at *τ* = 2 ns will decrease by 22.9% and 23.4% in the easy-cone and pMTJ, respectively. However, as shown in Figure 3b, the *J*_sw0_ of the pMTJ changes by 56% and this value in the easy-cone MTJ is only 8%. This indicates that although *J*_sw0_ of the pMTJ has been reduced a lot at elevated *T*, the required *J* for an appropriate write error rate cannot benefit from it. In contrast, *J* in the easy-cone MTJ becomes much smaller when *T* is increased. This demonstrates that the easy-cone MTJ exhibits a better performance at high temperatures. In addition, it can be observed that the slope of the write error rate curves in both devices is almost unchanged as *T* is varied. This is attributed to the opposite impact of *M*_sat_ and *P* on the write error rate slope. When *T* is raised, both *M*_sat_ and *P* are reduced, and it has been shown that a smaller *M*_sat_ leads to a larger slope, whereas a reduced *P* makes the curve more gradual [40]. Since the slope of write error rate curve remains the same, the effect of different *T* is manifested in the shift of the curves, which is much larger in the pMTJ due to the large variations of *J*/*J*_sw0_ required for a specific write error rate. In contrast, there is negligible shifts in the easy-cone MTJ, which is beneficial when the device is required to work stably in a wide temperature range.

## 4. Conclusions

In conclusion, we used the macrospin model to investigate the temperature dependence of the writing performance of the easy-cone MTJ and pMTJ. Consistent with the previous studies, at the same temperature, easy-one MTJ has a better writing performance [13,17,24]. In addition, the investigation of the temperature effect revealed that the writing performance of the easy-cone MTJ varies little under temperature variations, leading to its superiority in operating at high temperatures. We first examined the intrinsic switching current density and switching delay. *J*_sw0_ and *t*_sw_ of both devices were decreased at high temperatures, and for the pMTJ, they fell even faster. With the effective thermal field introduced, we then performed the stochastic simulations. In the fast precession regime, the easy-cone MTJ and pMTJ had an opposite temperature dependence in the current overdrive *J*_sw_/*J*_sw0_. At higher temperatures, easy-cone MTJ had a smaller *J*_sw_/*J*_sw0_. Meanwhile, the write error rate curve shifted in opposite directions with increased temperatures for the two devices. It was also demonstrated that the required current density for an appropriate write error rate is smaller in the easy-cone MTJ. Our work reveals the potential of the easy-cone MTJ-based STT-MRAM in commercial electronic devices. Benefiting from its outstanding performance at high temperatures, the easy-cone MTJ is also fully capable of working in the automotive IC.

## Figures and Tables

**Figure 1 nanomaterials-13-00337-f001:**
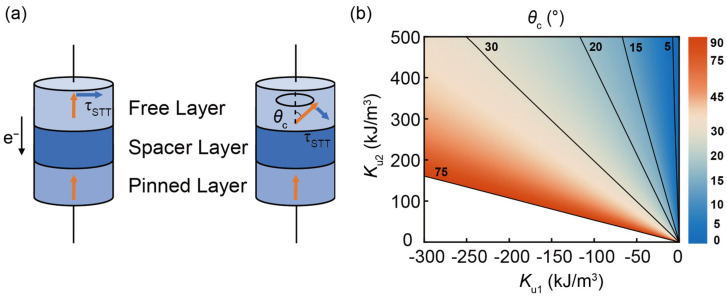
(**a**) Schematic of magnetic tunneling junctions with perpendicular and tilted magnetized FL. The orange arrows represent the magnetization of FL and PL. The blue arrows represent the spin transfer torque exerted on the magnetization. *θ*_c_ is defined as the polar angle of easy-cone FL magnetization. (**b**) *θ*_c_ as a function of K_u1_ and K_u2_.

**Figure 2 nanomaterials-13-00337-f002:**
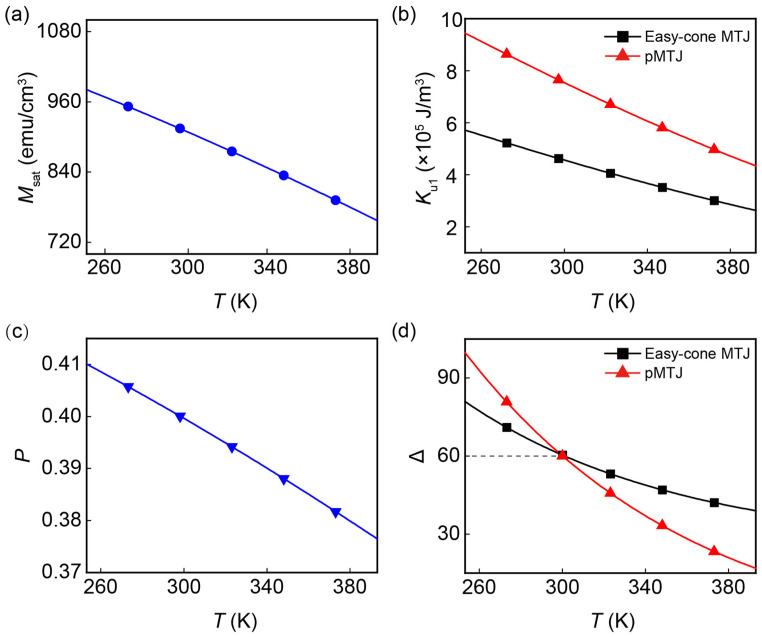
Temperature dependence of physical parameters of MTJ device: (**a**) saturation magnetization *M_sat_*, (**b**) first order anisotropy constant *K_u_*_1_, (**c**) spin polarization factor *P*, and (**d**) thermal stability Δ. Both devices have Δ = 60 at room temperature.

**Figure 3 nanomaterials-13-00337-f003:**
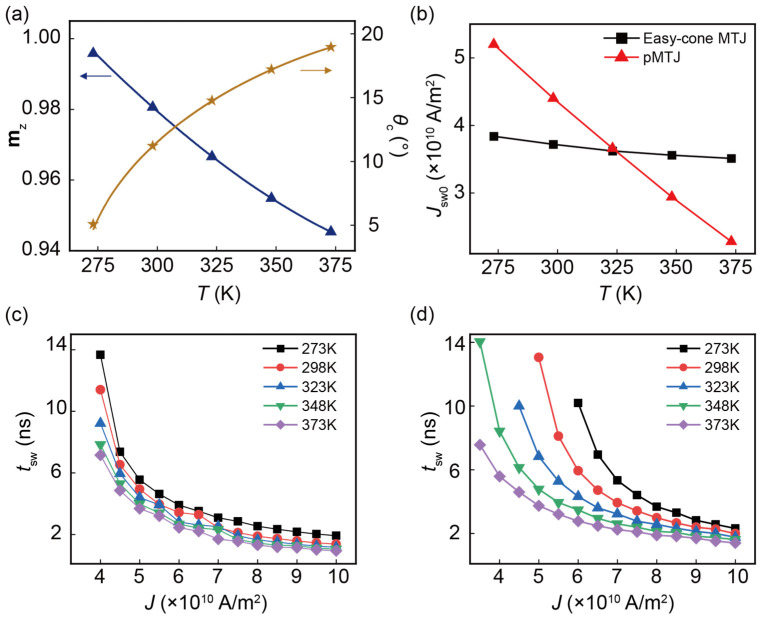
(**a**) The equilibrium **m**_z_ and initial polar angle *θ*_c_ as a function of temperature. (**b**) Temperature dependence of the intrinsic switching current density *J*_sw0_ in the easy-cone MTJ and pMTJ. Switching delay *t*_sw_ as a function of current density in the (**c**) easy-cone MTJ and (**d**) pMTJ. *T* = 273 K, 298 K, 323 K, 348 K, and 373 K are represented by square, circle, triangle, down-pointing triangle, and rhombus, respectively.

**Figure 4 nanomaterials-13-00337-f004:**
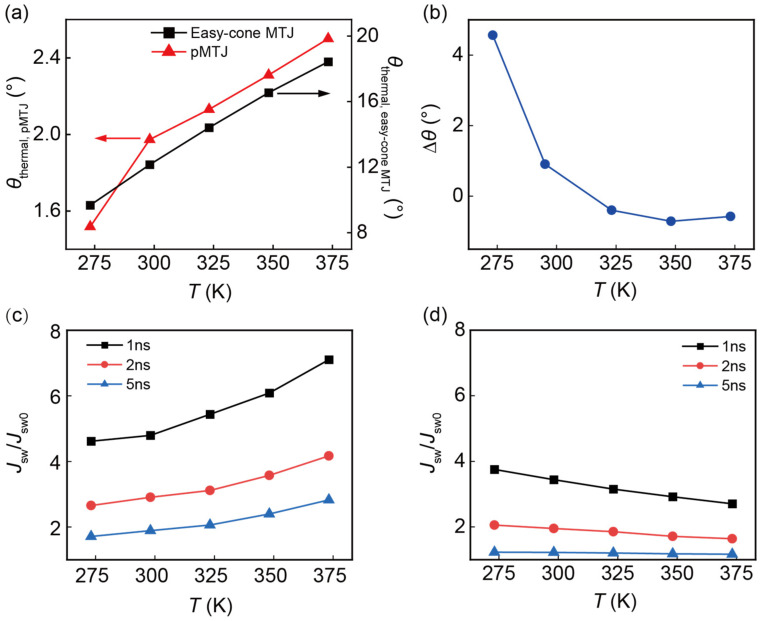
(**a**) *θ*_thermal_ as a function of temperature for the easy-cone MTJ and pMTJ. (**b**) The difference between *θ*_c_ and *θ*_thermal_ is represented as Δ*θ*. The current overdrive as a function of temperature in (**c**) pMTJ and (**d**) easy-cone MTJ.

**Figure 5 nanomaterials-13-00337-f005:**
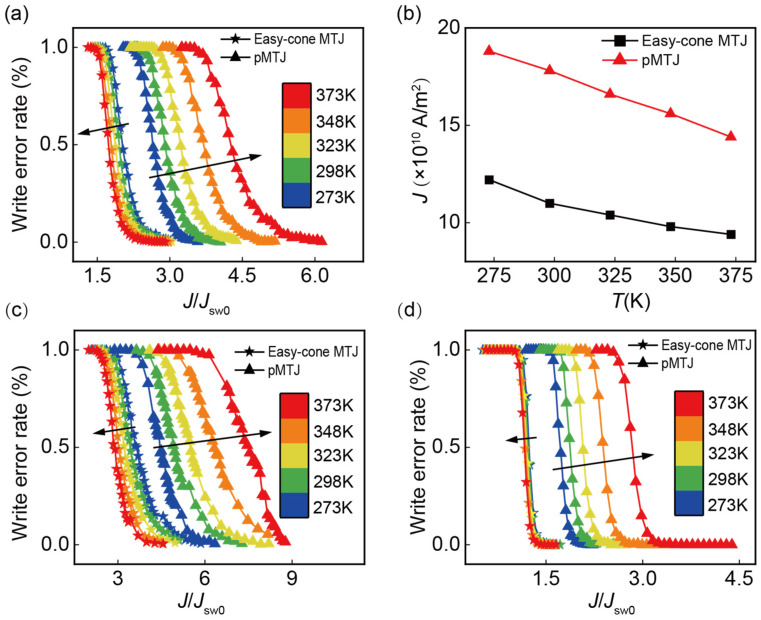
(**a**) Write error rate as a function of current density (normalized by *J*_sw0_) with pulse duration *τ* = 2 ns. The easy-cone MTJ and pMTJ are represented by star and triangle, respectively. The arrow indicates the direction of increased temperature. (**b**) The required current density for the write error rate of 10^−3^ at *τ* = 2 ns. Write error rate as a function of current density (normalized by *J*_sw0_) with pulse duration *τ* of (**c**) 1 ns and (**d**) 5 ns.

**Table 1 nanomaterials-13-00337-t001:** Parameters of the system.

Parameter	Unit	Value
*P*(0)	~	0.446
*K*_u1_(0) for the pMTJ	J/m^3^	1.82 × 10^6^
*K*_u1_(0) for the easy-cone MTJ	J/m^3^	1.1 × 10^6^
*M*_sat_(0)	A/m	1.22 × 10^6^
*t* _FL_	nm	1.2
*l* _FL_	nm	48
*w* _FL_	nm	20

## Data Availability

Data is available at the correspondence author.

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
