# Peer review of "Comparative Study of Temperature Impact in Spin-Torque Switched Perpendicular and Easy-Cone MTJs"

_nanomaterials, 2023, doi:10.3390/nano13020337_

Round 1
Reviewer 1 Report
Authors study about 2 types of spin transfer torque operated magnetic tunnel junctions (MTJs) by means of numerical simulations.
They focused on the temperature effect for pMTJ and easy-cone MTJ on their properties such as switching current density, switching deray time, and write error rate which are important for practical memory use.
As results, typically significantly lower write error rate of easy-cone MTJ compared with pMTJ at high temperature is clearly demonstrated, which is advantageous for stable operation of memory devices employing easy-cone MTJ. This study shows potential of easy-cone MTJ and attracts interest from people in spintronics.
The points of the manuscript are clear, and the main text is written in clear and correct English.
Overall quality of the manuscript for physics is sufficient and basically suitable for publication.
But it still has some small points should be addressed before publication as follows:
1) In line 5, The "and" should be before the last author.
2) In line 143, the caption for Table1 should be revised.
3) In line 126, "K" used for Boltzmann constant should be small letter.
4) For figure 4 (c) and (d), same axis range is desired to be reader friendly.
Reviewer 2 Report
In this article, authors presented macrospin model to investigate the effect of temperature on writing performance of easy-cone MTJ and pMTJ. The results are interesting. I have specific points which they might consider to improve this work.
1. Is there any relevant work been done in the past? From the survey, i figured out the temperature effects are being studied before and similar findings were reported. What is new in this work? Authors should mention significance of this work.
2. The authors used only modelling to investigate the effects of temperatures on writing performance of MTJ. How about they fabricate a model device and test experimentally these effects and demonstrate some basic results. In this way this work will be more technically sound.
3. I would like to see the device structures here, atleast schematic diagrams can fulfil the requirements.
4. Why they only use narrow temperature range of 273-373 K, they could have prolong this range too.
5. What is usually temperature requirements for STT-MRAMs?
6. The English standard of this article needs to be improved.
Reviewer 3 Report
Manuscript: nanomaterials-2115816
The manuscript titled "Comparative study of temperature impact in spin-torque switched perpendicular and easy-cone MTJs" by Jingwei Long et al present results on the use of macrospin model to investigate the effect of temperature on the writing performance of the easy-cone MTJ and pMTJ. I have following observations:
1// What is the basis to use macrospin model to perform such investigations. Please add other model(s) in introduction section to justify the same.
2// Please also include information for the temperature region where the blackbody radiation starts to dominate.
3// I am still not sure that how authors could only determine temperature region, 275-375 K, for performing such calculations. What will be the effect for lower temperatures < 275 K and higher temperatures >375 K.
4// Line 292-293 (This indicates that the easy-cone MTJ has a wider operating temperature range): Then I wonder why authors did not perform calculations in lower or higher regions.
5// Figure 2(a) says saturating magnetization. However, I do not see any saturation in the curve. They are more linear lines with the negative slope.
6// The motivation and the performed calculations are unclear in the present state. Authors need to work a lot on the objectives of the paper to make it to the standards of "Nanomaterials".
7// There should be a comparison of the results with existing literature in the last section. This will help the potential reader to understand the positives and negatives of microspin model and its applicability.
Based on these, I recommend a Major revision for the present manuscript.
Round 2
Reviewer 2 Report
This work can be accepted now.
Reviewer 3 Report
The manuscript is revised and can be accepted for publication now.